# The Role of US in Depicting Axillary Metastasis in High-Risk Breast Cancer Patients

**DOI:** 10.3390/jpm11121379

**Published:** 2021-12-16

**Authors:** Roxana Pintican, Magdalena Maria Duma, Madalina Szep, Diana Feier, Dan Eniu, Iulian Goidescu, Angelica Chiorean

**Affiliations:** 1Department of Radiology and Medical Imaging, Iuliu Hațieganu University of Medicine and Pharmacy, 400000 Cluj-Napoca, Romania; madalinaszep@gmail.com (M.S.); diana.feier@gmail.com (D.F.); chiorean_angi@yahoo.com (A.C.); 2Medimages Breast Center, 400458 Cluj-Napoca, Romania; magdaduma@gmail.com; 3Department of Gynecology, County Emergency Hospital, 400000 Cluj-Napoca, Romania; daneniu@yahoo.com (D.E.); iuliangoidescu@gmail.com (I.G.)

**Keywords:** high-risk breast cancer, axillary metastasis, ultrasound, BRCA, *ATM*, CHECK, *PALB*

## Abstract

Purpose: The aim of this study is to evaluate the role of US in depicting axillary nodal disease in high-risk patients with and without pathogenic mutations. *Methods:* The retrospective study included consecutive high-risk breast cancer (BC) patients who underwent a multigene testing panel for hereditary cancers, pre-operative axillary US and breast/axillary surgery. The group was divided into patients with pathogenic mutations (PM group) and patients without PM. Statistical analyses were performed using GraphPad Prism by applying Chi-square and Fisher exact tests, with a reference *p*-value < 0.05 and a CI of 95%. *Results:* Out of 190 patients with BC, 96 (51%) were negative and 94 (49%) were positive for PM as follows: 28 (25.5%) *BRCA1*, 16 (17%) *BRCA2*, 15 (16%) CHECK2, 14 (14%) RAD Group, 7 (7%) *PALB*, 6 (6%) *NBN*, 3 (3%) *TP53* and *ATM* and 2 (2%) *BARD1*. US was positive in 88 of the patients, 36 with PM and 52 without PM. US and surgery (≥N1 stage) were both positive in 31 (62%) of PM patients and 44 (88%) of patients without genetic changes. There were 19 (61%) false negative US examinations in the PM group and 6 (13%) in the group without genetic changes, respectively. If the US is positive, there is a 2.6 times greater risk of positive nodes in PM patients (*p*-value < 0.000, 95% CI = 4.2–37.9), and a 6.2 times greater risk of positive nodes in patients without genetic changes (*p*-value < 0.000, 95%CI = 8.4–37.4). In the PM group, US compared to surgery reached a sensitivity = 62, with PPV = 86 and NPV = 67. In the *BRCA1*/*2* subgroup, there is 2.5 greater times risk of nodal disease if the US is positive (*p*-value = 0.001, 95%CI = 2.6–76). In patients without PM, US compared to surgery reached a sensitivity = 88, PPV = 84 and NPV = 86. *Conclusion:* US is more sensitive in depicting axillary nodal disease in high-risk patients without PM compared to PM patients. Furthermore, there are more false negative US examinations in PM patients, compared to surgery patients.

## 1. Introduction

Axillary nodal disease is an important factor in the staging, management and prognosis of breast cancer (BC) patients. The majority of early BC patients will undergo a sentinel lymph node biopsy (SLNB) in order to determine the axillary lymph node status. The ACOSOG trial showed that, in cases with limited SLNB metastasis, axillary lymph-node dissection (ALND) can be omitted with equal overall survival for patients receiving breast conservative surgery [1,2], suggesting that ultrasound (US) might gain importance in the pre-therapeutic quantification of axillary tumor burden.

Recent guidelines recommend US as a part of the pre-therapeutic evaluation of early BC patients [3,4,5], trying to stratify who will further benefit from SLNB or ALND. Despite the fact that axillary US is generally used in clinical routine, and multiple studies exist on this matter, no generally accepted or standardized criteria for lymph node positivity exist. Moreover, wide range, variable values were reported for sensitivity of 26%–94%, and for specificity of 53%–98% of axillary US [6,7,8,9]. 

Limited data are available regarding the role of axillary US in high-risk breast cancer patients. Studies suggest that patients with pathogenic mutations (such as *BRCA1*, *BRCA2*, *ATM* or *CHEK2*) are more prone to have axillary metastasis, related to their highly aggressive breast cancers [10,11,12]. However, no imaging modality was evaluated nor reported in identifying them. On the one hand, up to 50% of the mutation carrier patients will have positive SLNB. On the other hand, early BC patients of average and low risk will have extensive axillary disease after positive SLNB only in a minority of cases (between 5% and 10%). Moreover, up to one-third of them have less than three positive nodes, and 65% of them are restricted to a single node [13]. 

Considering that SLNB is performed with the aim of tumor staging and not as a tre*ATM*ent option, much attention has been given to the identification of an additional diagnostic imaging tool for axillary assessment.

Our purpose is to evaluate the role of US in depicting axillary nodal disease in patients with hereditary breast cancer with and without pathogenic mutations.

## 2. Results

Out of 190 patients with BC, 51% were negative and 49% were positive for pathogenic mutations, with the percentages as follows: 29.8% *BRCA1*, 17% *BRCA2*, 16% CHECK2, 14% RAD Group (*RAD51C* and *RAD51D*), 7% *PALB*, 6% *NBN*, 3% *TP53* and *ATM* and 2% *BARD1*. Figure 1

The majority of breast cancers were invasive ductal carcinoma of “no special type” (IDC-NST), with triple negative (TN) and luminal B as the most frequently encountered molecular subtypes in the carrier group (30 and 33, respectively), and luminal A and triple-negative subtypes in the non-carrier group (46 and 27, respectively). 

There were 42 N0 patients in the carrier group and 46 N0 in the non-carrier group. A total of 52 patients were found to have axillary metastasis (N1, N2 or N3) in the carrier group and 50 patients in the non-carrier group, respectively. The majority of patients from both groups had early-stage BC, with 49 carrier patients and 52 non-carrier patients with less than stage IIa of the disease at the time of diagnosis (Table 1).

We analyzed the nodal status and pathology findings and found that positive N stage corelated with a breast tumor size larger than 2 cm and ki67% proliferative index > 20% in the carrier group with a high tumoral grade and ki67% proliferative index > 20% in the non-carrier group, respectively. The negative estrogen receptor (ER) status did not correlate with either group (Table 2).

On imaging, US was positive in 88 of the patients, 36 carriers and 52 non-carriers (Figure 1). US and surgery (≥N1 stage) were both positive in 31 (62%) of the carrier and 44 (88%) of the non-carrier patients. The axillary US was a false negative in the identification of suspect lymph nodes in 19 (38%) carrier patients with pathology-proven metastasis and in 6 (12%) negative, non-carrier patients, respectively (Table 3, Figure 2 and Figure 3).

In terms of diagnostic accuracy, US reached a sensitivity of 62% and a PPV of 86% in the carrier group, while in the non-carrier group, the sensitivity was higher at 88% (*p*-value = 0.00). In *BRCA1* and *BRCA2* subgroup, the US sensitivity was 62% with a specificity of 89%. The overall study group US sensitivity was 75% with a PPV of 85% (Table 4).

## 3. Discussion

In the present study, we aimed to assess the role of US in depicting axillary nodal disease in a high-risk BC population, with and without pathogenic mutations.

We found that carrier patients of *BRCA1, BRCA2, ATM, PALB, CHEK* and *TP53* mutations have 61% false negative axillary US examination, compared to only 13% false negative examinations in the non-carrier patients. This observation could have several explanations. First, the hypothesis should be related to their aggressive histology and behavior, with high tumoral grade, high proliferative index and high proportion of ER negative tumors leading to an increased risk of axillary micrometastasis. A micrometastasis is defined as a small tumoral deposit between 0.2 and 2 mm found within a lymph node and is easily misdiagnosed on US. In our cohort, only one *BRCA1* patient had one lymph node with micrometastasis, classified as N1 (N1 mi in the pathology report), which could explain the false negative US appearance. Second, the lack of a standardized criteria to classify “abnormal” lymph nodes on US may represent a bias in the selection process. We considered diffuse and focal cortical thickness > 3 mm, round/irregular shape and absent hilum as indicative US features for malignancy. The majority of published papers examined only the ipsilateral axilla and use a cut-off of 4 mm cortical thickness as abnormal. In our study, patients with ipsilateral diffuse cortical thickness > 3 mm were also evaluated on the contralateral axilla. Our diagnostic confidence increased when contralateral nodes showed a thin cortex of 1–2 mm. Because the contralateral axilla was not assessed for all patients, statistical analysis could not be performed in the current study. However, assessing the contralateral axilla could improve the US diagnosis of lymph node with diffuse cortical thickness of only 3 mm. Additionally, taking into account other US findings such as increased hypoechoic cortex, increased cortical vascularity and stiff elastography appearance, could help in redefining US criteria of suspect lymph nodes with no more than 3 mm cortical thickness.

The overall study population diagnostic accuracy of US was in concordance with recently published papers on a large cohort of patients of average risk [2,14,15]. We obtained an overall sensitivity of 75% with a PPV of 85%, but a slightly lower sensitivity of 66% due to the pathogenic mutation carrier patients. The lowest sensitivity of 66% was obtained in the subgroup of *BRCA1* and *BRCA2* patients, but with a high specificity of 89%.

Our results suggest that axillary US is not accurate enough in identifying nodal metastasis in pathogenic mutation carrier patients, even compared to high-risk, non-carrier patients. Positive N stage was depicted in 38% of the carrier patients with a high proportion on false negative US results of up to 61%. For this particular group of patients, additional imaging diagnostic tools needs to be assessed and/or implemented. SLNB could be indicated in carrier patients with early BC, despite the US examination. Regarding the high-risk BC patients without pathogenic genetic mutations, US could still play a role in the stratification and further indication of SLNB. Of course, additional studies on a larger high-risk BC population need to be performed to validate our findings.

Several papers assessed factors that could contribute to the positive axillary metastasis. Large tumor size, high tumoral grade, proliferative index and negative ER status were reported to correlate with axillary metastasis [16,17,18,19]. We found that N stage correlates with tumor size and proliferative index in the carrier group, and with tumoral grade and proliferative index in the non-carrier group. Our findings highlight and support the idea of pathophysiology differences between tumor masses in patients with and without pathogenic genetic mutations.

Large breast tumors are associated with positive node status, and this can reflect the tumor proliferative capacity to some extent. An increase in tumor size by 1 mm increases the risk of metastasis by 1.048, even in early-stage BC with T1-2N0M0 patients [16]. However, only one study analyzed tumor size as a risk factor in *BRCA1/2* carriers and concluded that no association exists [10]. To the best of our knowledge, no data exist regarding the other mutations associated with an increased risk of developing BC. In our study, the carrier group consisted of 52% T1-2 breast tumors, while in the non-carrier group, 54% had T1-2 tumors. The positive axillary nodal disease was correlated to large breast cancer masses of >2 cm in the carrier group, but no correlation between tumor size and N status was found in the non-carrier group. This could be explained by the presence of *non-BRCA* mutation types in the carrier group and additional risk factors for patients with increased familial risk that still need to be identified.

In concordance to previously published papers, we observed that tumoral grade 3 and high proliferative index of > 20% are associated with an increased risk of axillary metastasis [16].

The negative ER status did not correlate with N stage in neither group. One study divided *BRCA1* and *BRCA2* patients in TN and non-TN patients and found no correlation between N stage, tumor size and ER. Furthermore, the same study reported increased OR for axillary involvement of both TN and non-TN patients according to tumor size in the control group negative for *BRCA* mutations [10]. This supports and further sustains the hypothesis of different pathophysiology and development of breast cancer in high-risk patients.

One particular study addressed the role of US in the evaluation of axilla in patients at high risk for axillary metastasis [20]. However, the description of the study population is lacking and does not mention if there were large breast tumors or patients with genetic changes. To the best of our knowledge, no published paper addressed the role of US in the identification axillary lymph nodes metastasis in high-risk BC patients. Our high-risk population consisted of patients with positive familial history (at least one first-degree relative with BC or two second-degree relatives with BC), half of them with pathogenic mutations.

Furthermore, the role of US may gain even more significance in the newer radiomics-based studies that aim to predict metastatic nodes by the means of texture analysis [21].

Our study has some limitations of note. First, because the experienced and beginner breast imager assessed the images together and reached a consensus, no inter-reader variability tests could be performed. Second, our study is retrospective and unicentric. Third, there is a relatively small sample size, which could be balanced by the rarity of the pathogenic mutation carriers. Of note, our study included patients with only one pathogenic variant; further studies should also address and analyze the infrequent cases of double heterozygosity in breast cancer susceptibility genes [22].

## 4. Material and Methods

This was a retrospective study performed between March 2018 and December 2019 on consecutive patients. Our ethics committee approved the study and waived a written consent from participants.

The inclusion criteria consisted of histology-proven BC patients with multigene panel test for hereditary cancers, pre-operative breast and axillary US, complete surgery, pathology and immunohistochemistry reports. We excluded patients who refused surgery (12) or had incomplete US images (5) or unavailable axillary pathology reports (3). Currently, the genetic testing is indicated in our institution if at least one of the following criteria are met: one first-degree relative with BC diagnosed < 50 years or two second-degree relatives with BC diagnosed < 50 years, age < 35 years at diagnosis, triple-negative subtype or bilateral breast cancer; and if additional melanoma, colon, pancreas or ovarian cancer is present in the patient’s personal history or that of their relatives. All included patients had hereditary breast cancer with or without pathogenic variants. The genetic testing was performed on DNA extracted from blood samples with further molecular analysis by the means of next generation sequencing (NGS) identifying five types of variants: benign, more likely benign, variants of uncertain significance (VUS), likely pathogenic and pathogenic.

Two US machines were used to assess the patients, a HI VISION Ascendus (Hitachi Ltd., Tokyo, Japan) and Supersonic Mach (Aixplorer Ltd., Aix-en-provence, France). One experienced breast imaging doctor with > 15 years of experience on breast imaging performed the US and review the images together with a breast junior radiologist (3 years of experience). The lymph nodes with at least one abnormal US feature were further classified as abnormal on imaging. The US features included cortical thickness ≥3 mm (if 3 mm was depicted, the contralateral axilla was assessed and compared), an eccentric cortical hypertrophy ≥3 mm, cortex microcalcifications, a round shape and the loss of central fatty hilum.

All performed axillary biopsies were US-guided core-needle biopsies. No fine needle aspiration was performed. Figure 4

Germline DNA for *BRCA1, BRCA2, CHEK2, RAD51C, RAD51D, PALB, NBN, TP53, ATM* and *BARD1* mutation testing was derived from blood samples.

Statistical analyses were performed using GraphPad Prism by applying Chi-square and Fisher exact tests, with a reference *p*-value < 0.05 and a CI 95%.

## 5. Conclusions

US is more sensitive in depicting axillary nodal disease in high-risk negative, non-carrier patients, compared to carrier patients.

Breast cancer patients with pathogenic mutations (such as *BRCA1*/*2*, CHECK, *PALB* and *ATM*) are more prone to have false negative axillary US examinations compared to high-risk breast cancer patients without genetic changes.

## Figures and Tables

**Figure 1 jpm-11-01379-f001:**
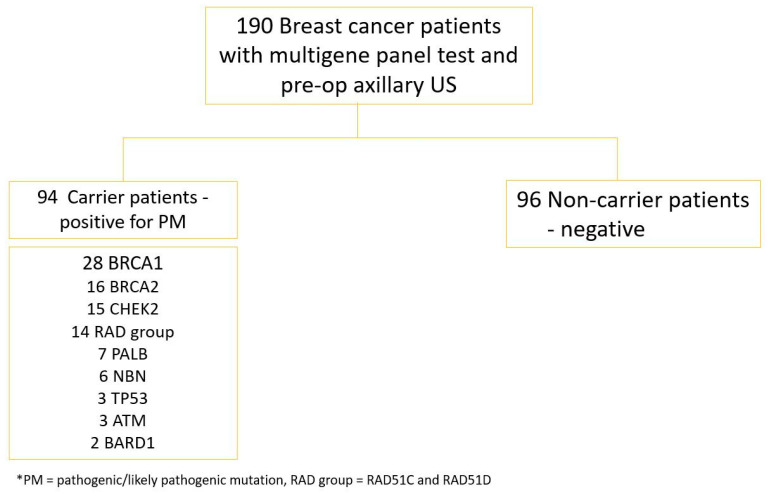
Study population.

**Figure 2 jpm-11-01379-f002:**
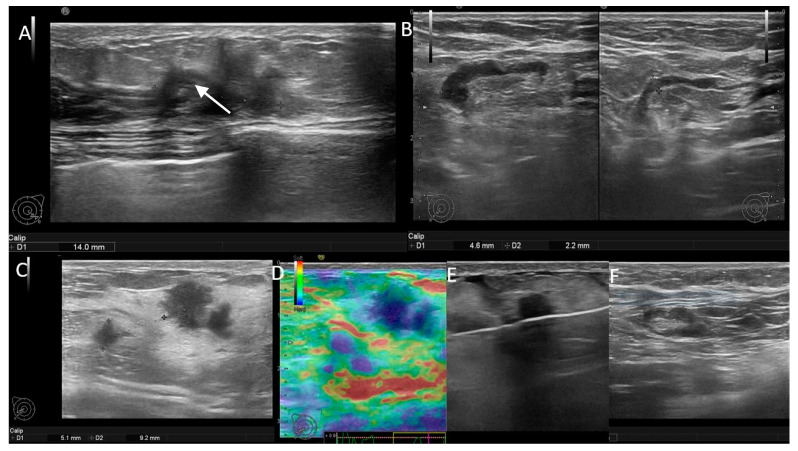
(**A**,**B**) Patient with negative genetic testin-unifocal BIRADS 5 mass and positive axillary US. There is an ipsilateral lymph node with cortical thickness up to 4.4 mm (white arrow), compared to the contralateral node, which has only 2.2 cortical thickness. (**C**–**F**) Patient with *BRCA2* mutation-bifocal BIRADS 5 masses and negative axillary US. There is an ipsilateral node with fatty hilum and a thin cortex of 2 mm.

**Figure 3 jpm-11-01379-f003:**
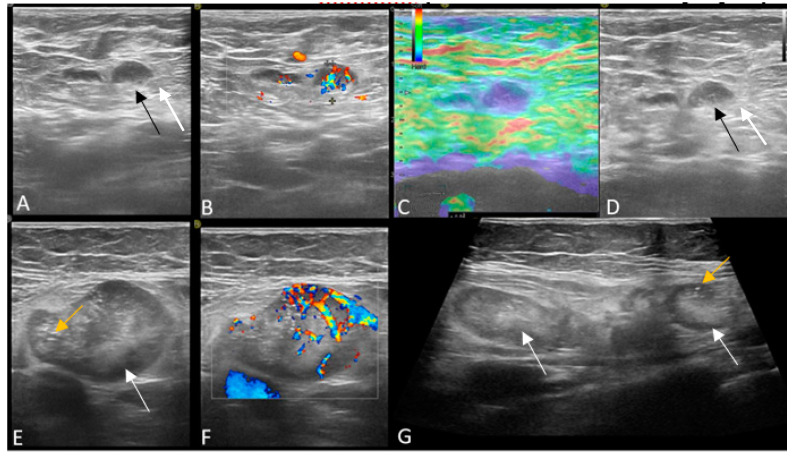
(**A**–**D**) Patient with negative genetic testing and positive axillary US. There is a lymph node with focal cortical thickening (white arrow), increased vascularity and stiff strain elastography appearance. (**E**–**G**) Patient with *CHEK2* mutation and positive axillary US. There are multiple, irregular lymph nodes, with eccentric hilum (white arrow), thickened cortex with punctate microcalcs (yellow arrow) and chaotic, periphery vascularity.

**Figure 4 jpm-11-01379-f004:**
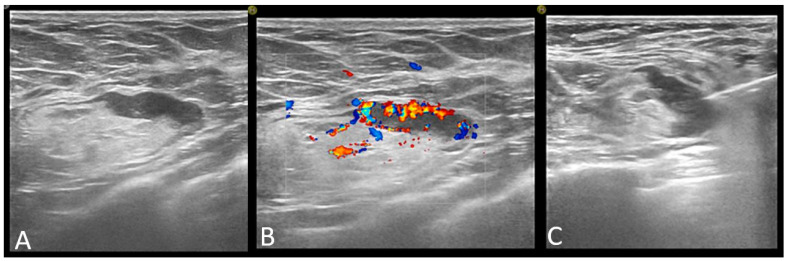
Axillary lymph node classified as abnormal on ultrasound: focal cortical thickening of more than 3 mm (**A**) with chaotic vessels (**B**), and core-needle biopsy was performed (**C**).

**Table 1 jpm-11-01379-t001:** Genetic testing and pathology results for study population.

		Pathogenic Mutations									Without PM *	
Age (mean)		24–81 (45.26)									30–79 (45.3)	
Genetic alterations		*BRCA1*	*BRCA2*	CHECK2	*PALB*2	RAD group	*NBN*	*ATM*	*TP53*	*BARD1*		
		28	16	15	7	14	6	3	3	2		
Breast cancer type	IDC NST ^1^ * otherwise specified	1 medullary 2 papillary					1 tubular				1 adenoid cystic 4 medullary 3 mucinous 3 tubular	
Molecular type	Luminal A	0	4	4	2	4	2	1	0	0	46	
	Luminal B	6	12	6	4	0	2	2	0	1	23	
	HER 2 enriched	1	0	5	1	5	2	0	0	0	0	
	Triple negative	21	0	0	0	5	0	0	3	1	27	

N status	N0	16	5	9	1	5	3	1	0	2	46	
	N1	8	10	2	4	2	1	0	3	0	23	
	N2	4	1	4	2	4	2	2	0	0	24	
	N3	0	0	0	0	3	0	0	0	0	3	

Total Nr of	N0				42						46	
	> N1				52						50	

Stage of disease	Early Stage I-IIa	19	4	10	2	6	3	3	0	2	52	
	Advanced > Stage IIa	9	12	5	5	8	3	2	3	0	44	

Total Nr of patients						94					96	

^1^ IDC NST = invasive ductal carcinoma no special type; * PM = patients tested for pathogenic mutations for *BRCA1*/2, CHECK2, *PALB*2, RAD group, *NBN*, *ATM*, *TP53*, *BARD1*.

**Table 2 jpm-11-01379-t002:** Nodal status and pathology findings in study population.

		Pathogenic Mutations	Without Pathogenic Mutations
Axillary US	+	36	52
	-	58	44
Surgery	+	50	50
	-	44	46
True positive		31	44
True negative		39	38
False negative		19	6
False positive		5	8
Total nr of patients		94	96
PPV		86%, 95% CI 0.72–0.94	85%, 95%CI 0.78–0.90
NPV		67%, 95% CI 0.59–0.72	75%, 95%CI 0.69–0.80

US = ultrasound; + = positive (for ultrasound, positive = suspicious features: round shape, absent hilum, cortical thickness >3 mm); - = negative; PPV= positive predictive value; NPV = negative predictive value; Mentioning that true and false positives are reported regarding the axillary US.

**Table 3 jpm-11-01379-t003:** Axillary nodal status and immunohistochemistry and histology findings.

Pathogenic Mutations								
	Breast tumor size		Estrogen receptor		Tumoral grade		Ki67% proliferative index	
	<2 cm	≥2 cm	+	-	G1	G2 + G3	>20%	<20%
N0	29	14	27	15	3	39	26	26
≥N1	23	29	39	13	1	51	47	5
*p*-value	0.038		0.26		0.32		0.001	
**Without Pathogenic Mutations**								
	Breast tumor size		Estrogen receptor		Tumoral grade		Ki67% proliferative index	
	<2 cm	≥2 cm	+	-	G1	G2 + G3	>20%	<20%
N0	29	18	35	12	21	26	19	28
≥N1	23	26	33	16	7	68	31	18
*p*-value	0.147		0.44		0.001		0.04	

True positive = patients with suspect US confirmed by surgery; true negative = patients with no metastatic lymph nodes confirmed by histology; false negative = patients with no suspect lymph nodes on US, but with surgery-proven metastasis; false positive = patients with abnormal US lymph nodes with no histology-proven metastasis.

**Table 4 jpm-11-01379-t004:** Se, SP, PPV and NPV of US in study population.

		Sensitivity	Specificity	PPV	NPV
Mutation status	Positive	62	88	86	67
	Negative	88	82	84	86
*BRCA1* and *BRCA2* subgroup		62	89	88	65
Overall study group		75	85	85	75

Numbers are %; *BRCA1* and *BRCA2* subgroup = patients with pathogenic variant in wither *BRCA1* or *BRCA2* gene; NPV = negative predictive value; PPV = positive predictive value. Overall study group = mutation-positive plus mutation-negative patients.

## Data Availability

All data is available at the corresponding author.

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
