# Peer review of "The Role of US in Depicting Axillary Metastasis in High-Risk Breast Cancer Patients"

_jpm, 2021, doi:10.3390/jpm11121379_

Round 1

Reviewer 1 Report

In the present manuscript authors analyzed the role of ultrasound in depicting axillary metastasis in patients with high-risk breast cancer with and without pathogenic mutations.The manuscript is very interesting and can be accepted for publication if the authors are ready to incorporate the following revisions:

- The authors should specify better that the enrolled patients are patients with hereditary breast cancer, with and without mutations, but all selected according to HBOC selection criteria.

- At line 71 the percentage of mutation in the BRCA1 gene appears to be wrong, it should be 29.8%. Please to correct.

- At line 77-79  the authors should specify the number of patients which report the various specific subtypes (TN, luminal B, etc).

- In Table 2 it should be indicate the total number of patients with and without pathogenic mutations and the total number of N0 patients  and >= N1.

- At line 108 the authors should specify how the 61% and 13% values were obtained, that appears unclear in text and in table. It should be better explained both in text and in table.

- At line 109 should be described what is shown in figure 3.

- In the text the reference to figure 4 is not shown, please to correct.

- In table 4 the “BRCA1 + BRCA2” seems to refer to patients who report both mutations, but in text it does not seem to be indicated as such. Please clarify this aspect. In addition, is unclear why are reported only BRCA mutated patients and not patients with mutations in other genes.

- In material and methods the authors should indicate the method which the mutation analysis was performed.

- The authors describe patients with a single mutation, they  observed cases of patients reporting a double mutation in the same or two genes, or better or worse prognoses, US associated, have been described in patients with a double mutation?

- The authors would benefit from reading the following articles, the contents of which could be useful for improving the manuscript:

  1. PMID: 32315268 DOI: 10.1148/radiol.2020192534
  2. PMID: 32329142 DOI: 10.1002/jum.15294
  3. PMID: 23940062 DOI: 10.1515/cclm-2013-0263

Reviewer 2 Report

The authors examined the role of US in depicting axillary node metastasis of breast cancer according to the status of pathogenic mutation.

Although some data is interesting, the manuscript is too descriptive.

  1. The authors described that “if the US is positive, there is a 2.6 times greater…6.2 times grater risk…(lines 131-133)”. This is difficult of be understood. Is this an odds ratio? Please explain how the percentage was calculated.

  1. The authors described that false negative was 61% and 13% in carrier patients and non-carrier patients, respectively. However, according to table 3, the number of false negative 19 out of 50 (surgery +) (38%) in PM group and 6 out of 50 (12%). In general, false negative rate (%) = 1-sensitivity (%).

  1. The authors described that the difference of false negative rate is partially due to ER of Ki67 status. However, ER negative patients is 28 out of 94 in PM group while 28 out of 96 in non-PM group and almost comparable. In addition, as shown in table 2, ER status was not correlated with nodal status in both groups. How about calculating false negative rate according to Ki67 status?

Round 2

Reviewer 2 Report

The manuscript is well revised and suitable for publication.